# Online-Assisted Survey on Antibiotic Use by Pet Owners in Dogs and Cats

**DOI:** 10.3390/antibiotics13050382

**Published:** 2024-04-24

**Authors:** Clara Rocholl, Yury Zablotski, Bianka Schulz

**Affiliations:** Clinic of Small Animal Medicine, Centre for Clinical Veterinary Medicine, Ludwig Maximilian University of Munich, 80539 Munich, Germany

**Keywords:** antimicrobial, resistance, canine, feline, CIA, therapy, veterinary, antibiogram, fluoroquinolones, cephalosporins

## Abstract

The aim of the study was two-fold: first, to collect data on the use of antibiotics in Germany for dogs and cats and, second, their owners’ experiences and opinions. Using an anonymous online survey, dog and cat owners were asked about the last antibiotic administration in their pet. The inclusion criterion was any antibiotic administration within the last year. A total of 708 questionnaires from 463 dogs and 245 cats could be evaluated. Diarrhea was reported as the most common reason for antibiotic administration in dogs (18.4%). Wound infection/abscess/bite injury was the second most common reason in dogs (16.0%). In cats wound infection/abscess/bite injury was the most common reason (23.3%), followed by dental treatment (21.2%) and upper respiratory tract infections (16.7%). The most common antibiotics used systemically in both species were amoxicillin/clavulanic acid (32.5%), amoxicillin (14.8%), metronidazole (6.9%), and doxycycline (6.8%). While efficacy (99.9%) and tolerability (94.8%) were rated as most important for the choice of antibiotics, costs (51.6%) were cited as predominantly unimportant. First-line antibiotics were used significantly more often than critically important antibiotics. The majority of animal owners show awareness for avoidance of antibiotic resistance and the use of critically important antibiotics.

## 1. Introduction

According to the World Health Organisation (WHO), rising antimicrobial resistance (AMR) is a threat to human and animal health [1]. In 2019, there were 1.27 million deaths worldwide due to AMR [2]. In Germany, there were approximately 2400 AMR-related deaths recorded in 2015 [3]. 

The emergence of AMR is a natural process that can be accelerated by the improper use of antibiotics [4]. Antibiotic therapy should be given only for bacterial infections and with clear indication of the appropriate agent and suitable dosage, duration, and mode of application. To improve antibiotic efficacy and thus minimize the development of resistance, it is important to reduce overall consumption, improve the use of diagnostic tests, use critically important agents (CIA) prudently, and optimize dosing regimens [5]. The use of antibiotics can also lead to the development of resistances in pets, and moreover, the transmission of resistant pathogens to humans and vice versa [6,7,8]. The close coexistence with pet dogs and cats can also increase the risk for humans to become infected with resistant pathogens [9].

In Germany, a total of 15.7 million cats were kept in 25% of all households and 10.5 million dogs were kept in 21% of households in 2023 [10]. In 2000, the German Federal Veterinary Surgeons’ Association (BTK = Bundestierärztekammer) published the “Guidelines for the prudent use of veterinary antimicrobial drugs” [11]. To combat rising AMR, the Veterinary In-House Dispensaries Ordinance (Verordnung über tierärztliche Hausapotheken, TÄHAV) was renewed in 2018. Because of these guidelines, a ban on reclassification (§ 12b) and an antibiogram requirement (§ 12c) were introduced for the use of 3rd and 4th generation fluoroquinolones and cephalosporins [12]. These drug classes are considered the “highest priority critically important antimicrobials” (HPCIA) for human medicine according to the WHO’s 2019 assessment [13]. These antibiotic classes have also been classified as “critically important antimicrobial agents” (VCIA) in veterinary medicine [14]. 

To date, hardly any data regarding the exact amount of antibiotics used in dogs and cats exists in Germany. The German Federal Office of Consumer Protection and Food Safety (BVL) has been evaluating the quantities of antibiotics dispensed by pharmaceutical companies and wholesalers to German veterinarians since 2011 [15]. However, no distinction has been made between the different animal species when recording antibiotic dispensing volumes [15]. With the EU Regulation 2019/06, enforced since 28 January 2022, data of the applied drugs with antibacterial effect must be reported annually. For “other animals”, including dogs and cats, data collection is a specified goal starting in 2030 (Regulation (EU) 2019/6 Article 75 paragraph 5). In overall antibiotic dispensing, recorded by the BVL, a significant decrease has been recorded from 2011 to 2021, for 3rd/4th generation cephalosporins and fluoroquinolones. A continuous decline in the quantities of antibiotics that were dispensed was recorded in 2022 [15,16]. Mohr and co-workers compared the use of antibiotics in dogs and cats reported by veterinarians in Bavaria, before and after the amendment of the TÄHAV using an anonymous online survey [17]. By comparing the dispensing volumes published by the BVL, Mohr and co-workers showed a significantly decreased usage of 3rd and 4th generation cephalosporins and fluoroquinolones in 2020 compared to before the amendment in 2017 [17]. Two other surveys among veterinarians in Berlin and across Germany also showed a positive impact of the amendment, with respondents reporting less frequent use of HPCIA and more frequent implementation of antibiograms [18,19].

Smith and co-workers reported that veterinarians commonly feel pressured to prescribe antibiotics because pet owners pay for treatment, and therefore, tend to expect some kind of active therapy [20]. In contrast, owners often felt that veterinarians prescribed antibiotics too frequently [20]. They reported that they would not be disappointed if veterinarians did not give an antibiotic and would rather follow their alternative recommendation [20]. Redding and co-workers similarly showed that most owners trusted their veterinarian regarding antibiotic administration [21]. On the other hand, the majority of pet owners stated that they would prefer to treat their pet with antimicrobial drugs even if the benefit of an antibiotic was not clear [21].

When choosing an antibiotic, German veterinarians ranked sensitivity followed by ease of administration as important, while cost was cited as rather unimportant [22]. A particularly easy-to-administer antibiotic licensed for dogs and cats is Convenia^®^ (active ingredients: cefovecin, Zoetis, Germany), which is effective for up to 14 days after a single injection by the veterinarian. The inability to administer oral medications to the cat is the most frequently cited reason for cefovecin administration among UK cat owners, in one study [23]. Surveys among cat owners in the UK and owners of dogs and cats in North America have shown that a single long-acting injection is preferred over oral medication [24,25,26]. In the UK 29.6% of participating owners reported their cat received a long-acting injection when given antibiotics [26].

The aim of the study was to collect data from dog and cat owners in Germany on antibiotic administration, including questions on the type of antibiotic, form of application, tests carried out beforehand, and factors influencing antibiotic prescription. Of special interest was the usage reported for the 3rd generation cephalosporine, cefovecin, as a critically important antibiotic.

## 2. Results

A total of 709 questionnaires were completed. One questionnaire was excluded because it was stated that it came from Austria. Thus, 708 questionnaires were evaluated, of which 463 (65.4%) were completed for a dog and 245 (34.6%) for a cat.

### 2.1. Pet Owner Demographics

The majority of participants were female (676/705; 95.9%) and the average age (median) was 39 years. In total, 29.3% (206/704) of pet owners reported having medical training. Of these owners, 45.4% (93/205) reported having training in veterinary medicine and 54.6% (112/205) in human medicine. Among the participants with veterinary education, 23 had a university degree, and among those with human medical education, 24 had a university degree. Overall, 28.5% (201/706) of participants had a college degree. The complete data can be viewed in the Appendix A.

### 2.2. Demographic Data of Dogs and Cats

One hundred and fifteen of the dogs were female intact (24.8%) and 120 female neutered (25.9%), 111 were male intact (24.0%) and 117 male neutered (25.3%). Twenty-nine cats were female intact (11.8%), 18 were male intact (7.3%), 85 female neutered (34.7%), and 113 male neutered (46.1%). The median total age was 5 years for both dogs (IQR 3–9) and cats (IQR 3–10) between 1 and 18 years of age (*p* = 0.4). An age of less than one year was reported in 24 dogs (5.2%) and in 12 cats (4.9%). An age of 20 years or more was indicated in 3 cats (1.2%). The majority of dogs (333/462; 72.1%) and cats (210/239; 87.9%) were purebred, and owners of 105 cats indicated that their pet was an outdoor cat (42.9%). Most participants reported that their pet did not have pet health insurance (534/708; 75.4%). Dogs were significantly more likely to have health insurance (134/463; 28.9%) than cats (40/245; 16.3%) (*p* < 0.001). The complete data can be viewed in the Appendix A.

### 2.3. Antibiotic Administration

#### 2.3.1. Indications for Antibiotic Administration

An overview on indications for antibiotic prescription is shown in Table 1.

In dogs and cats, the most frequently cited reason was wound infection/abscess/bite injury (131/708; 18.5%), followed by diarrhea (107/708; 15.1%), dental treatment (90/708; 12.7%), and urinary tract infection (74/708; 10.5%). Diarrhea was the most frequently reported reason in dogs (85/463; 18.4%) and was significantly more frequently reported in dogs than in cats (22/245; 9.0%) (*p* = 0.003). Wound infection/abscess/bite injury was the most frequently cited reason in cats (57/245; 23.3%) and the second most frequently cited reason in dogs (74/463; 16.0%) (*p* = 0.047). Subsequently, dental treatment (52/245; 21.2%) and upper respiratory tract infection (41/245; 16.7%) were documented as the second and third most common reasons in cats. In addition, both reasons were reported significantly more frequently in cats than in dogs (*p* < 0.001 in each case). The majority of owners (580/708; 81.9%) reported expecting an antibiotic prescription for their pet’s problem when presenting their pet to a veterinarian. There was no significant difference regarding the expectation of antibiotic treatment between participants with (176/206; 85.4%) and without (400/498; 80.3%) medical training (*p* = 0.109).

The owner expectations for antibiotic administration for different indications are displayed in the Appendix A. The owners’ highest expectation of antibiotic administration was for castration (40/42; 95.2%), soft tissue surgery (64/69; 92.8%), and other surgery (11/12; 91.7%). 

#### 2.3.2. Application Form

Antibiotics were most commonly administered in the form of tablets (581/708; 82.1%), followed by injections (182/708; 25.7%), topical administration (44/708; 6.2%), oral administration of solutions/suspensions/pastes (38/708; 5.4%), and capsules (8/708; 1.1%). The majority of injections were short-acting (128/158; 81.0%). Administration of long-acting injections were reported in 19.0% (30/158). Overall, there was a significant difference in the types of injection between dogs and cats (*p* < 0.001). Short-acting injections were reported more frequently in dogs (98/96; 92.7%) than in cats (39/62; 62.9%). Long-acting injections were reported more frequently in cats (23/62; 37.1%) than in dogs (7/96; 7.3%) (Table 2)

#### 2.3.3. Duration and Frequency of Antibiotic Administration 

A total of 86.7% (549/633) of pets were administered an antibiotic orally or locally between 1 and 14 days with a median of 7 days. An administration of 15 days or longer was reported by 13.3% (84/633) of participants. Most participants indicated that the antibiotic should have been given twice a day (388/625; 62.1%). A once-daily administration was reported by 32.5% (203/625) of the pet owners. Only 29 participants reported administering the antibiotic 3 times per day (4.6%). Administration four times per day was reported by 2 individuals (0.3%), and administrations 5 times, 6 times, or more than 6 times daily was reported by 1 participant each (0.2%). The doses given more than four times a day were topical treatments. (Appendix A)

Long-acting injections were predominantly administered once (17/30; 56.7%), followed by injections twice (12/30; 40.0%), and three times (1/30; 3.3%). More frequent administration was not reported for long-acting injections. A single injection was also reported by most owners for short-acting injections (60/125; 48.0%). A two-time application was reported by 19.2% (24/125), and 12.8% (16/125) reported a three-time administration. In addition, 22 participants reported administration between 4 and 10 times (22/125; 17.6%), and 2.4% (3/125) reported administration more than 10 times. There was no significant difference in the number of injections given between long- and short-acting injections (*p* = 0.315). The complete data can be viewed in the Appendix A.

#### 2.3.4. Administered Antibiotics

Of the systemically applied antibiotics, amoxicillin/clavulanic acid was used most frequently (221/708; 32.5%), followed by amoxicillin (101/708; 14.8%), metronidazole (47/708; 6.9%), and doxycycline (46/708; 6.8%). Cefovecin as a 3rd generation cephalosporin was administered significantly more frequently in cats (12/245; 5.0%) than in dogs (1/463; 0.2%) (*p* < 0.001). A total of 35.5% (242/708) of pet owners did not remember which antibiotic they had administered (Table 3). An overview on the selection of systemic antibiotics in regard to the disease process can be found in Appendix A.

Local antibiotic therapy was reported in a total of 44 cases. Due to the small number of cases, a statistical comparison between dogs and cats was not performed. The three most commonly used local antibiotics were polymyxin (9/44; 20.5%), gentamicin (7/44; 15.9%), and chloramphenicol (6/44; 13.6%). Ten owners did not know which topical antibiotic had been administered (22.7%) (Table 4)

#### 2.3.5. Pet Owner Education by the Veterinarian and Pet Owner Compliance

We surveyed dog and cat owners using a 5-point Likert scale regarding the education provided by the vet and compliance with antibiotic administration. Data are provided in the Appendix A.

With regard to compliance, the majority stated that they followed the veterinarian’s recommendation exactly when administering the antibiotics (37/699; 5.3% agreed and 647/699; 92.6% strongly agreed). Accordingly, most owners reported sticking to the exact number of tablets/capsules (12/582; 2.1% agreed and 567/582; 97.4% strongly agreed) and to the exact amount of solution/suspension/paste (0/38 agreed and 37/38; 97.4% strongly agreed) per antibiotic administration. Similarly, the majority reported adhering to the exact number of antibiotic administrations per day (16/640; 2.5% agreed and 618/640; 96.6% strongly agreed) and the exact time intervals between administrations (186/636; 29.2% agreed and 412/636; 64.8% strongly agreed). Figure 1 shows the questions and the respective results of the Likert scale.

Participants with medical training were significantly more likely to report that they had been involved and advised by their veterinarian in the decision of the choice of the form of administration (*p* = 0.007), and that they had been informed about possible side effects (*p* < 0.001). For all other items, there was no significant difference between owners with and without medical training (Appendix A).

#### 2.3.6. Feasibility of Antibiotic Administration

The ease of antibiotic administration was reported as being significantly better in dog- compared to cat owners (*p* < 0.001). In addition, significantly more cat owners reported to barely being able to administer the medication orally or not at all (*p* < 0.001). In contrast, there was no significant difference between cat and dog owners for application of local therapy (*p* = 0.089) (Table 5).

Most owners administered the tablets or capsules hidden in food or treats (264/584; 45.2%), there was no significant difference for the application form between dogs and cats (*p* = 0.105). Details are provided in Table 6.

About half the respondents (336/600; 56.0%) stated that they washed their hands after antibiotic administration. This was not routinely done by 247 owners (41.2%), 4 participants (0.7%) did not wash their hands but wore gloves and thirteen (2.2%) pet owners both washed their hands and wore gloves. A significant difference was found between participants with medical training and those without (*p* = 0.014). Of the participants with medical training, 62.4% (111/178) washed their hands and 33.1% (59/178) did not. Among those without medical training, 53.3% (224/420) washed their hands and 44.5% (187/420) did not wash their hands. In addition, participants with medical training were more likely to wear gloves. The full data are available in the Appendix A.

#### 2.3.7. Adverse Reactions and Premature Discontinuation of Administration

Termination of antibiotic administration before the full duration instructed by the veterinarian was reported by only 4.1% (29/706) of the owners. The most frequently cited reason was the occurrence of side effects (9/29; 31.0%), followed by consultation with the veterinarian (8/29; 27.6%), difficulty with administration or use (7/29; 24.1%), fear of possible side effects (2/29; 6.9%), and recovery of the dog or cat (2/29; 6.9%).

Side effects that were attributed to antibiotic usage were reported by 21.9% of owners (145/662). Diarrhea was the most frequently mentioned side effect (87/145; 60.0%), followed by vomiting (25/145; 17.2%), and allergic reactions (25/145; 17.2%). In addition, 35.2% (51/145) reported that their animal had experienced other side effects.

Multiple answers were possible (Appendix A).

#### 2.3.8. Culture and Sensitivity Testing before Antibiotic Administration

About half (364/708; 51.4%) of pet owners stated that no tests had been performed before antibiotic administration. In total, 19.3% of owners (125/648) indicated that culture and sensitivity testing (C&S) or other tests for pathogen detection had been performed (details in the Appendix A). C&S was significantly more often performed before systemic administration of fluoroquinolones (enrofloxacin, marbofloxacin, pradofloxacin) and 3rd generation cephalosporins (cefovecin) (21/42; 50.0%) compared to other antibiotics (85/430; 19.8%) (*p* < 0.001) (Appendix A). C&S was most frequently performed for urinary tract infections (27/125; 21.6%), wound infection/abscess/bite injury (21/125; 16.8%), diarrhea (20/125; 16.0%), and ear infections (17/125; 13.6%) (Appendix A). 

### 2.4. Preferred Route of Administration for Systemic Antibiotic Administration

Overall, tablet administration was most preferred by both dog (119/463; 25.7% strong and 277/463; 59.8% very strong) and cat (56/245; 22.9% strong and 94/245; 38.4% very strong) owners. The owners’ ratings are shown in Figure 2 and the complete data can be viewed in the Appendix A.

### 2.5. Tablet Administration

Cat owners were significantly more likely to report problems with tablet insertion (*p* < 0.001). Cats (81/216; 37.5%) more frequently accepted liquid/paste medications over tablets/capsules compared to dogs (73/360; 20.3%) (*p* < 0.001). In addition, cat owners (107/177; 60.5%) more commonly indicated that tablet intake was related to taste (*p* = 0.003) than dog owners (182/390; 46.7%) (detailed data shown in Appendix A).

### 2.6. Choosing an Antibiotic

Efficacy was ranked the most important factor by both dog and cat owners (dog 6.7% very important and 93.1% extremely important; cat 5.3% very important and 94.7% extremely important, *p* = 0.684), followed by tolerability (potential side effects) (dog 21.0% very important and 75.6% extremely important; cat 25.3% very important and 66.1% extremely important; *p* = 0.004). Costs were cited as least important in both species, but dog owners found it significantly more important (dog 13.2% very important and 7.3% extremely important; cat 10.2% very important and 2.9% extremely important; *p* = 0.007). The owners’ ratings are shown in Figure 3 and the complete data can be viewed in the Appendix A. 

## 3. Discussion

One aim of the present study was to collect data on common indications for antibiotic therapy in dogs and cats in Germany. Diarrhea was the most frequently cited reason for antibiotic administration in dogs in the present study. This is consistent with previous studies that also identified gastrointestinal or intestinal disease as a frequent indication for antibiotic administration [28,29]. However, current studies and guidelines state that there is no indication for the usage of antibiotics in acute uncomplicated diarrhea without evidence of systemic inflammation or sepsis [30,31,32,33,34,35,36]. Studies have examined the effect of antibiotic administration in acute diarrhea, and it was found that the duration to clinical improvement did not differ with and without the administration of amoxicillin/clavulanic acid or metronidazole [31,37]. 

Wound infections, bite injuries and abscesses were the most common reasons for antibiotic treatment in cats and the second most common indication in dogs in the present survey. However, detailed data on extent, location and severity of the condition could not be obtained for the large number of cases, to assess the justification for antibiotic treatment for the pets of participating owners. According to the FECAVA (Federation of European Companion Animal Veterinary Associations) guidelines, uncomplicated skin lesions or mildly infected wounds and bite wounds can frequently be treated with local antiseptic therapy only [34]. Because abscesses are encapsulated, systemic antibiotic therapy may be inefficient due to failure of antibiotics to penetrate the capsule wall, resulting in insufficient drug concentrations reaching the target area [38]. For superficial abscesses, local therapy using drainage, irrigation, and debridement may often be sufficient [32,34,35,36,38]. Deeper and penetrating bite wounds as well as extensive tissue damage can be indications for antibiotic therapy [32]. 

Antibiotic administration was reported for dental treatments by 21.2% of cat owners and 8.2% of dog owners. In the case of dental prophylaxis, tooth extractions or infections of the oral cavity such as periodontitis, stomatitis, gingivitis or even periapical tooth root abscess, the administration of systemic antibiotics is usually not indicated, instead tooth cleaning and, depending on the severity, tooth extraction are advised [32,34,35,36,39,40]. Exceptions representing indications for systemic antibiotic administration include, for example, signs of systemic infection, immunosuppression, or severe metabolic, or cardiac disease [32,33,36].

Cats were significantly more likely to receive an antibiotic for upper respiratory tract infections and rhinitis. Feline rhinitis is mostly viral in origin, but secondary bacterial infections are commonly present. Antibiotic administration is indicated in cats with mucopurulent discharge in the presence of fever, inappetence, or lethargy [41]. In cases of chronic upper respiratory tract infection with a duration of more than 10 days, bacteria are considered secondary pathogens and diagnostic work-up of primary underlying conditions is recommended instead [32,33,41].

Urinary tract infections represent another prevalent indication for antibiotic administration in the present study. In the presence of clinical signs, bacterial urinary tract infection can be diagnosed in 43–65% of dogs and only 2–19% of cats, therefore making antibiotic treatment commonly unnecessary despite suspicious clinical signs in feline patients [42,43]. Overall, feline idiopathic cystitis is more common in cats than bacterial infection [44,45]. Therefore, urinalysis, including bacterial culture of cystocentesis urine, should always be performed prior to administration of an antibiotic, especially in cats [46].

Castrations were listed less frequently as a reason for antibiotic therapy, with a total of 5.9%. When comparing the expectation to receive an antibiotic with the reason for administration, it is noticeable that most owners expected an antibiotic for castration (95.2%). Since castration is an elective and clean procedure and the risk of postoperative infection is low, no preoperative or perioperative antibiotic administration is recommended for healthy patients [32,34,36]. 

It is a striking result of the study that 81.9% of owners were expecting antibiotic therapy for their pet. In contrast, in a study in the UK, only 49.2% of cat owners expected antibiotics for treatment [26]. 

Oral antibiotic administration was reported most frequently in this study (tablets, solutions/suspensions/pastes or capsules), followed by injections and topical application. In other studies, oral antibiotic administration was also the most common type of medication, followed by injections and topical therapy [26,47]. Tablets were the most commonly given oral formulation, followed by oral solutions/suspensions/pastes, and capsules, similar to what has been described by cat owners in the UK [26]. The second most common form of antibiotic treatment in this study was by injection, with a total of 25.7%. In contrast to dogs, cats received a long-acting formulation more frequently, but more commonly short-acting antibiotics. This is in contrast to a survey performed in the UK in 2019, in that cats were more likely to receive a long-acting injection (29.6%) rather than a short-acting injectable preparation (11.7%) [26].

The most commonly used systemically applied antibiotics in our survey were amoxicillin/clavulanic acid (32.5%), amoxicillin (14.8%), metronidazole (6.9%), and doxycycline (6.8%). These findings are comparable to the results of another German investigation among veterinary practitioners revealing amoxicillin/clavulanic acid, amoxicillin, and metronidazole as the most commonly used antibiotics in 2020 [17]. That study showed a significant increase in the use of amoxicillin, amoxicillin/clavulanic acid, doxycycline, and metronidazole in 2020 compared to 2017, while at the same time a significant decrease in fluoroquinolones and cefovecin over the same time period [17]. Penicillins were also the most commonly used antibiotics in dogs and/or cats in other studies [7,18,19,47,48,49]. This distribution pattern of antibiotic dispensing quantities can also be confirmed with the official data published by the BVL as the legal authority in Germany. Data from 2022 show that penicillins were the most frequently used antibiotics in veterinary medicine [16]. 

Surprisingly, metronidazole was indicated as the third most frequently used antibiotic in the present study. Metronidazole is a nitroimidazole and, unlike amoxicillin and amoxicillin/clavulanic acid, has a fairly narrow spectrum of activity against obligate anaerobes and protozoa like Giardia [50,51]. According to a study from the USA, metronidazole was the most commonly used antibiotic for gastrointestinal disorders in dogs and cats [29]. In the present study, diarrhea was also reported to be the most common reason for the administration of metronidazole. However, this should be discussed critically, as metronidazole is known to cause significant and longstanding dysbiosis of the enteral microbiome [52], and in most cases of acute and chronic diarrhea, antibiotic treatment does not affect clinical improvement in stable patients [30,31,32,33,34,35,36,37,53,54]. For treatment of Giardiasis, fenbendazole has been recommended as the first-line drug, causing only minimal changes in the enteral microbiota while being equally effective against Giardia [32,36,55,56,57]. 

The investigation revealed little usage of systemic fluoroquinolones and cefovecin as the only licensed 3rd generation cephalosporin. The low numbers for these drug classes may be due to legal restrictions implemented by the amendment of the TÄHAV in 2018 [12]. In 2018, a decrease in the dispensed amounts of fluoroquinolones and 3rd and 4th generation cephalosporins could already be observed in the published dispensing endings of the BVL, and in the most recent record of 2022, the lowest amount of dispensed antimicrobials of these classes since 2011 was recorded [16]. Thus, the investigation among owners confirms the data provided by the government and by surveys interviewing veterinarians throughout Germany showing a significant decrease in the use of these restricted antibiotic classes [17,18,19].

Overall, for most antibiotic classes there was no significant difference regarding prescriptions between dogs and cats. The exception was cefovecin, that with 5.0%, was used significantly more often in cats compared to 0.2% in dogs. Since cefovecin is a long-acting injection and 9.4% of cat owners indicated their animal had received a long-acting injection, the number of cefovecin administrations is probably higher than just 5.0%. Among dog owners, 1.5% reported long-acting injections. 

Other studies also showed a more frequent use of cefovecin in cats than in dogs [48,49,58,59]. A study comparing antibiotic use in different European countries showed significant differences for cefovecin usage in cats between Italy (50%), the Netherlands (0%), and Belgium (16%) [7]. Studies from the UK, Australia, and Canada published fractions of 17–32% in these countries for cats in contrast to 1–4% in dogs [26,48,49,59]. While studies from the UK, Australia, and Canada indicated frequent use of cefovecin especially in cats, data derived from this and a further study indicates lower, more prudent use of this last resort antibiotic in Germany [26,47,48,49,59]. In a German study, regular usage was reported by 20% of the veterinarians surveyed in 2021, most frequently in outdoor cats [19]. According to a study performed in the UK, the most common reasons for cefovecin administration were inability to administer oral medication and antibiotic treatment of stray cats [23]. Results from the present study also suggest that cat owners are significantly more likely to have problems with the administration of oral medications than dog owners, explaining the higher rate for cefovecin use in cats. 

Due to the legal obligation to perform resistance testing when using fluoroquinolones and cephalosporins of the 3rd and 4th generation in Germany, veterinarians are obligated to prepare an antibiogram before prescription. In the present study, owners indicated that an antibiogram was performed in 50% of cases before administrations of HPCIA. Significantly lower was the number of C&S before cefovecin use in studies from Australia (0.3%) and UK (0.4%) [23,58]. Similarly, no prior testing was reported in 61.3% of antibiotic administrations in the UK [26]. Overall, 80.7% of owners in the present survey reported antibiotic treatment without microbiological examination beforehand. Since about half of the owners (51.4%) stated that no tests were performed before antibiotic administration, it appears that these antibiotics were prescribed only on the basis of anamnestic information and clinical examination. Fever as a possible indication of systemic infection was given as a reason by only 6.6%. 

In the present survey, cat owners more frequently reported problems with oral medications than dog owners. In this survey, most cat owners administered the tablets along with food or hidden in treats, or by administration directly into the mouth. Overall, both dog and cat owners indicated that liquid/paste medications were not easier to administer than solid medications. Pleasant taste of the medication improves the ability to administer the drug according to 60.5% of cat owners and 46.7% of dog owners. Other studies from the UK described a better ability to enter liquids/pastes compared to tablets in cats [60,61]. In the present survey, tablets were found to be the preferred form of administration by dog and cat owners, followed by long-acting injections in cats and capsules in dogs, and in third position, solutions/suspensions/capsules in cats and long-acting injections in dogs. Overall, long-acting injections were always preferred over short-acting and thus usually multiple injections. In contrast, other studies showed that a single long-acting injection was preferred over tablet administrations by dog and cat owners [24,25,26]. In one study this was preferred, even if a long course of antibiotics was not regarded as necessary [26]. 

To increase owner compliance with antibiotic administration, it seems important to choose a feasible form of administration that does not cause negative reactions [62]. Our data underscores the importance of involving the owner in planning of the treatment protocol and educating them about administration of the medication. A study revealed that the feeling that the veterinarian has taken enough time for the treatment also increases the compliance of the pet owners [63]. 

In the present survey, 50.1% of owners reported having been involved in the choice of the application form; however, 15.8% wanted more education regarding antibiotic administration. Similarly, in a study performed in the UK 51.1% of owners were involved in the choice of application while 44.2% would have liked more training by the veterinarian on administration of medications [26]. Overall, owners demonstrated very good compliance in the present survey, with 97.9% reporting that they followed the veterinarian’s recommendations exactly. Similarly, compliance regarding dosage, number of administrations and exact time interval was also high in the present study with more than 94%. In contrast, other studies have shown lower compliance regarding dosage (91% and 84%, respectively), and treatment intervals (64% and 34%, respectively) [64,65]. While one study showed 9-fold better compliance with once- or twice-daily antibiotic administration compared to three times daily administration, no significant difference was found by two other studies [63,64,65]. However, studies in humans have also shown that compliance decreases with increasing daily dosages [66,67,68]. Since in the present study the majority of owners (94.6%) administered the antibiotic one or two times daily, this may have positively influenced overall compliance.

For the selection of antibiotics, factors such as efficacy, tolerability, prevention of antibiotic resistance, and veterinary recommendation were more important to dog and cat owners than administration method and palatability of the medication. Costs were cited overall as the least important influencing factor. However, it was noticeable that cat owners were significantly more concerned with ease of administration and palatability, whereas dog owners were significantly more concerned with tolerability and costs. In a recent study from Australia, effectiveness was also considered more important to owners than ease of administration or costs; however, 55% of pet owners indicated that treatment should be as cheap as possible [69]. In contrast to the present investigation in Germany, in North America, dog owners mentioned costs as the most important factor (47%), followed by route of administration (31%) and relevance of the drug to human medicine (22%); while among cat owners, costs (37%) and route of administration (38%) were seen as relatively equally important, while relevance to human medicine was rated lower with 21% [24,25]. The majority of UK cat owners indicated they would also be happy to pay for diagnostics to find out the most effective antibiotic for treatment of their pet [26]. Similar to results of the present study among owners, German veterinarians also indicated sensitivity as the most important influencing factor, followed by the administration method while costs were rated as rather unimportant [22]. Results indicated that the influence of costs might depend significantly on economic background in different countries.

Results of the present study show that many owners want to be involved and informed in the decision making of antibiotic prescriptions, especially regarding administration options and palatability. Antibiotics should be used in accordance with existing guidelines, and achieving good compliance by the owners is an important factor to assure correct and prudent use of antimicrobial drugs. Due to the high expectations of owners regarding the administration of antibiotics, owner education about prudent use of antimicrobials seems critically important. If further diagnostics are indicated to make a treatment decision, this should be discussed with the owners to assure an optimal selection process. After all, costs were the least important factor, while efficacy and avoidance of resistance problems were the most important factors to owners in the present study for antibiotic selection. HPCIA should be avoided and therefore not be given for convenience, for example, as a long-acting injection without a clear indication. Tablets were indicated as the preferred form of antibiotic administration by cat owners in our survey, but since cat owners have problems with administration more often, it is important to thoroughly educate owners on how to administer tablets and what aids are available. 

A limitation of an owner-based survey is the potential for false answers out of lack of knowledge or because owners did not remember details correctly. To reduce this risk, only owners who had administered antibiotics within the last year were included. Furthermore, there is the possibility of a selection bias, that more owners with an interest in the topic of antibiotics and resistance participated in the survey. This could have influenced the questions regarding compliance, among other things, in the direction of correct answers. Some of the participants were also recruited through dog and cat groups on social media, which may suggest an increased interest in the topic. It is also notable that most participants were female (95.9%). This is also comparable to other studies [26,69] and might have influenced the results. Another factor potentially influencing the results is that 29.3% of participating owners reported having received medical training. In addition, the design of the questions might have influenced the answers of owners, using phrases such as ‘always’, ‘strictly’ or ‘exactly’, and mentioning the problems of antibiotic resistance, that could potentially have promoted answers owners appreciated as being more socially desirable. 

Another limiting factor is that only general terms such as “diarrhea” were offered as possible answers regarding the reason for antibiotic administration. However, we know that diarrhea can have different causes and manifestations and can therefore also require different treatment approaches. In addition, no further details regarding the health status of the animal, concomitant diseases, performed diagnostics and diagnosis could be recorded. Thus, it is not possible to adequately decide on the indication for antibiotic administration. Further research would be indicated in this regard in order to assess more clearly the factors influencing inadequate antibiotic prescription and the influence of pet owners.

## 4. Material and Methods

The study was approved by the ethical committee of the veterinary faculty of LMU Munich (AZ 265-30-04-2021). By means of an anonymous online survey, dog and cat owners were asked about their experiences with the last antibiotic administration within the last year. The questionnaire was created with the program EvaSys and was accessible online from November 2021 to July 2022. It consisted of seven parts: (1) demographics of the dog or cat (2) information on last antibiotic administration (3) tests performed before antibiotic use (4) preferred route of administration (5) chosen antibiotic (6) demographics of owners. Except for the last section, all questions were mandatory and had to be answered in order to proceed with the questionnaire. Different question types were used such as single-choice, multiple-choice, and Likert scales. If necessary, there was the option “I do not know/unknown” for the pet owners. Five questions offered the opportunity to enter a free text, should none of the given answer options fit. Where possible, the answers to the open-ended questions were incorporated into the existing questions. In addition, the questionnaire consisted of several filter questions, to be able to adapt individual questions depending on the previous answer. The questionnaire translated into English is available in the Appendix A. 

Inclusion criteria for participation in the survey were that owners had given their dog or cat an antibiotic within the last year and that they lived in Germany. Veterinarians and people aged under 18 years were excluded. If owners had more than one dog or cat, they were asked to answer the questions for the animal that last received an antibiotic. A pre-test was performed with 50 dog and cat owners before publishing the questionnaire. The questionnaire was published on the website of the Clinic for Small Animal Medicine, LMU Munich, and a link to the survey was also distributed via social media (various dog and cat groups on Facebook and Instagram). In addition, a flyer was emailed to veterinary practices and clinics nationwide to be hung in waiting areas. Using the QR-Code in the flyer, owners could dial in directly to the survey.

### Statistical Analysis

All data were exported from EvaSys into Microsoft Excel (2016 MSO (16.0.4266.1001)). Statistical analyses were performed using R (4.2.0). All data have been analyzed descriptively, and statistical comparisons between groups in categorical variables have been performed using the Chi-squared test in samples > 5, while the Fisher’s exact test was used, if the theoretical frequency was less than five. Significance level was set at *p* < 0.05 for all comparisons and a correction for multiple testing was performed using Benjamini and Hochberg. The *p*-values stated in the results are corrected *p*-values and are described as q-values in the tables. Cramer’s V was used as the effect size for the association between two categorical variables and has values between 0 and 1. The larger the value of Cramer’s V, the greater the correlation between the values. In addition, the 95% confidence interval of the effect size was determined. When evaluating the reasons for antibiotic administration, the answer options “nasal discharge” and “infection of the upper respiratory tract“ were subsequently evaluated together as “infection of the upper respiratory tract “. Similarly, “cough” and “infection of the lower respiratory tract“ were combined under “infection of the lower respiratory tract”.

## Figures and Tables

**Figure 1 antibiotics-13-00382-f001:**
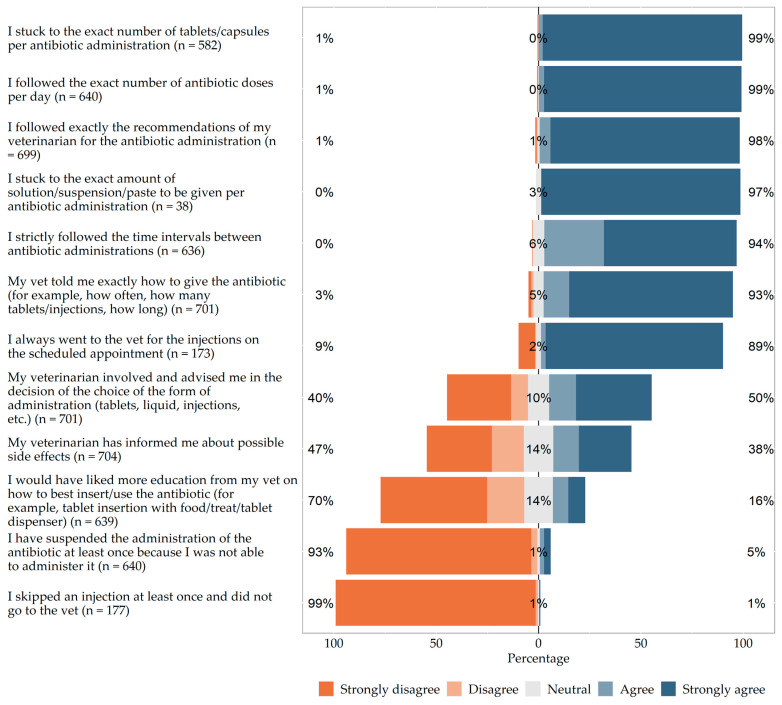
5-point Likert scale displaying veterinary education of pet owners and owner compliance with antibiotic administration.

**Figure 2 antibiotics-13-00382-f002:**
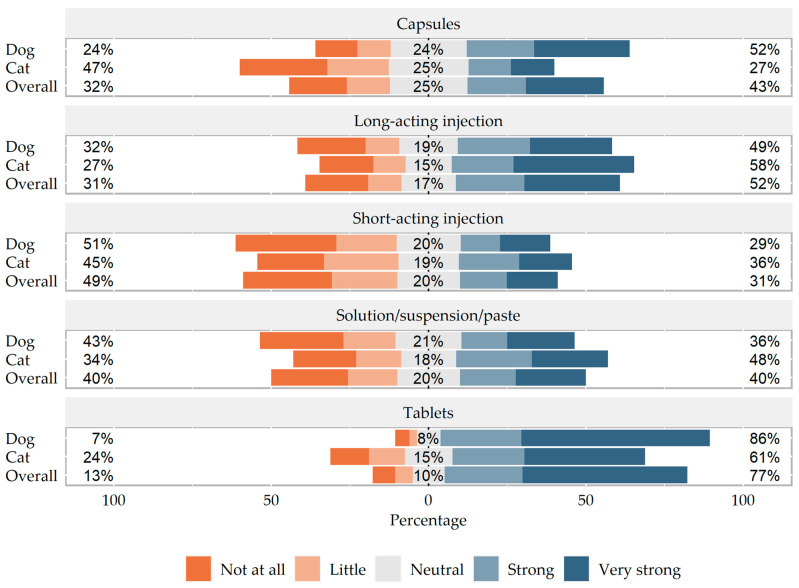
5-point Likert scale regarding preferred route for systemic antibiotic administration comparing dogs and cats.

**Figure 3 antibiotics-13-00382-f003:**
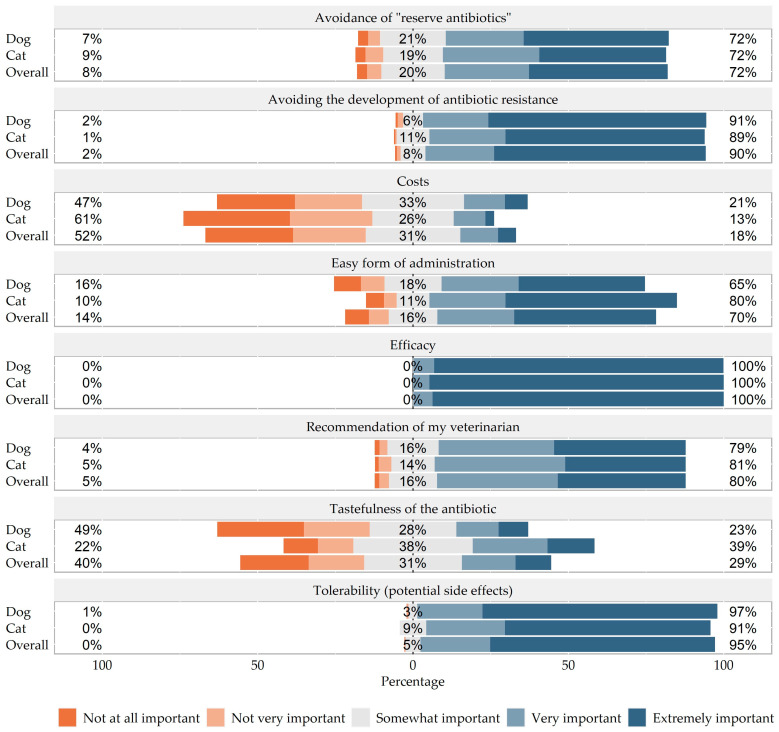
5-point Likert scale regarding influencing factors on antibiotic administration comparing dogs and cats.

**Table 1 antibiotics-13-00382-t001:** Reasons for the last antibiotic administration in dogs and cats and comparison between groups, multiple answers were possible.

	Overall n = 708	Dogs n = 463	Cats n = 245	*p*-Value ^1^	q-Value ^2^	Cramer’s V ^3^	95% CI ^4^
Wound infection/abscess/bite injury	131 (18.5%)	74 (16.0%)	57 (23.3%)	0.018	0.047	0.08	0.00–1.00
Diarrhea	107 (15.1%)	85 (18.4%)	22 (9.0%)	<0.001	0.003	0.12	0.05–1.00
Dental treatment	90 (12.7%)	38 (8.2%)	52 (21.2%)	<0.001	<0.001	0.18	0.12–1.00
Urinary tract infection	74 (10.5%)	49 (10.6%)	25 (10.2%)	0.875	0.875	0.00	0.00–1.00
Surgery on soft tissue	69 (9.7%)	52 (11.2%)	17 (6.9%)	0.067	0.107	0.06	0.00–1.00
Infection of the upper respiratory tract	63 (8.9%)	22 (4.8%)	41 (16.7%)	<0.001	<0.001	0.20	0.13–1.00
Vomiting	59 (8.3%)	45 (9.7%)	14 (5.7%)	0.067	0.107	0.06	0.00–1.00
Infection of the lower respiratory tract	54 (7.6%)	30 (6.5%)	24 (9.8%)	0.114	0.165	0.05	0.00–1.00
Skin problems	50 (7.1%)	37 (8.0%)	13 (5.3%)	0.185	0.246	0.03	0.00–1.00
Ear infection	50 (7.1%)	40 (8.6%)	10 (4.1%)	0.024	0.056	0.08	0.00–1.00
Fever	47 (6.6%)	27 (5.8%)	20 (8.2%)	0.236	0.290	0.02	0.00–1.00
Castration/neutering	42 (5.9%)	34 (7.3%)	8 (3.3%)	0.029	0.058	0.07	0.00–1.00
Eye problem	34 (4.8%)	21 (4.5%)	13 (5.3%)	0.648	0.741	0.00	0.00–1.00
Surgery of the bones	28 (4.0%)	27 (5.8%)	1 (0.4%)	<0.001	0.002	0.13	0.06–1.00
Other surgery	12 (1.7%)	7 (1.5%)	5 (2.0%)	0.761	0.811	0.00	0.00–1.00
Other problem	97 (13.7%)	82 (17.7%)	15 (6.1%)	<0.001	<0.001	0.16	0.09–1.00

^1^ Pearson’s Chi-squared test; Fisher’s exact test; ^2^ False discovery rate correction for multiple testing with Benjamini & Hochberg method; ^3^ Interpretation: >0.25 very strong, >0.15 strong, >0.10 moderate, >0.05 weak, >0 no or very weak association [27]; ^4^ CI = confidence interval of effect size.

**Table 2 antibiotics-13-00382-t002:** Application form of the last antibiotic administration in comparison for dogs and cats, multiple answers were possible.

Type of Administration and Formulation	Overall n = 708	Dogs n = 463	Cats n = 245	*p*-Value ^1^	q-Value ^2^	Cramer’s V ^3^	95% CI ^4^
**Tablets**	581 (82.1%)	402 (86.8%)	179 (73.1%)	<0.001	<0.001	0.17	0.10–1.00
**Injections**	182 (25.7%)	113 (24.4%)	69 (28.2%)	0.276	0.349	0.02	0.00–1.00
**Type of injection**				<0.001	<0.001	0.36	0.23–1.00
Short-acting	128/158 (81.0%)	89/96 (92.7%)	39/62 (62.9%)				
Long-acting	30/158 (19.0%)	7/96 (7.3%)	23/62 (37.1%)				
**Topical**	44 (6.2%)	32 (6.9%)	12 (4.9%)	0.291	0.349	0.01	0.00–1.00
**Oral solution/suspension/paste**	38 (5.4%)	8 (1.7%)	30 (12.2%)	<0.001	<0.001	0.22	0.16–1.00
**Capsules**	8 (1.1%)	5 (1.1%)	3 (1.2%)	>0.999	>0.999	0.00	0.00–1.00

^1^ Pearson’s Chi-squared test; Fisher’s exact test; ^2^ False discovery rate correction for multiple testing with Benjamini & Hochberg method; ^3^ Interpretation: >0.25 very strong, >0.15 strong, >0.10 moderate, >0.05 weak, >0 no or very weak association [27]; ^4^ CI = confidence interval of effect size.

**Table 3 antibiotics-13-00382-t003:** Systemically applied antibiotics in comparison for dogs and cats, multiple answers were possible.

	Overall n = 708	Dogs n = 463	Cats n = 245	*p*-Value ^1^	q-Value ^2^	Cramer’s V ^3^	95% CI ^4^
**Penicillins (aminopenicillins):**							
Amoxicillin	101 (14.8%)	66 (15.0%)	35 (14.6%)	0.893	>0.999	0.00	0.00–1.00
Ampicillin	2 (0.3%)	0 (0.0%)	2 (0.8%)	0.124	0.357	0.06	0.00–1.00
Penicillin	2 (0.3%)	2 (0.5%)	0 (0.0%)	0.543	0.769	0.01	0.00–1.00
**Penicillins (aminopenicillins with beta-lactamase inhibitors):**							
Amoxicillin/Clavulanic acid	221 (32.5%)	145 (32.9%)	76 (31.7%)	0.747	0.907	0.00	0.00–1.00
**Nitroimidazoles:**							
Metronidazole	47 (6.9%)	35 (7.9%)	12 (5.0%)	0.149	0.361	0.04	0.00–1.00
**Tetracyclines:**							
Doxycycline	46 (6.8%)	25 (5.7%)	21 (8.8%)	0.126	0.357	0.04	0.00–1.00
**Lincosamides:**							
Clindamycin	18 (2.6%)	9 (2.0%)	9 (3.8%)	0.184	0.391	0.03	0.00–1.00
**Fluorochinolones:**							
Enrofloxacin	18 (2.6%)	13 (2.9%)	5 (2.1%)	0.502	0.769	0.00	0.00–1.00
Marbofloxacin	15 (2.2%)	9 (2.0%)	6 (2.5%)	0.697	0.907	0.00	0.00–1.00
Pradofloxacin	2 (0.3%)	1 (0.2%)	1 (0.4%)	>0.999	>0.999	0.00	0.00–1.00
**Cephalosporins:**							
Cefovecin (3rd generation)	13 (1.9%)	1 (0.2%)	12 (5.0%)	<0.001	<0.001	0.16	0.10–1.00
Cefalexin (1st generation)	12 (1.8%)	12 (2.7%)	0 (0.0%)	0.011	0.091	0.09	0.00–1.00
Cefazolin (1st generation)	1 (0.1%)	0 (0.0%)	1 (0.4%)	0.352	0.599	0.04	0.00–1.00
**Trimethoprim and Sulfonamides** (TSO)	6 (0.9%)	6 (1.4%)	0 (0.0%)	0.095	0.357	0.06	0.00–1.00
**Aminoglycosides:**							
Gentamicin	1 (0.1%)	1 (0.2%)	0 (0.0%)	>0.999	>0.999	0.00	0.00–1.00
Other	7 (1.0%)	3 (0.7%)	4 (1.7%)	0.250	0.472	0.03	0.00–1.00
Unknown	242 (35.5%)	167 (37.9%)	75 (31.2%)	0.085	0.357	0.05	0.00–1.00

^1^ Fisher’s exact test; Pearson’s Chi-squared test; ^2^ False discovery rate correction for multiple testing with Benjamini & Hochberg method; ^3^ Interpretation: >0.25 very strong, >0.15 strong, >0.10 moderate, >0.05 weak, >0 no or very weak association [27]; ^4^ CI = confidence interval of effect size.

**Table 4 antibiotics-13-00382-t004:** Topically applied antibiotics in dogs and cats, multiple answers were possible.

	n = 44		n = 44
Polymyxin	9 (20.5%)	Fusidic acid	1 (2.3%)
Gentamicin	7 (15.9%)	Moxifloxacin	1 (2.3%)
Chloramphenicol	6 (13.6%)	Oxytetracycline	1 (2.3%)
Ofloxacin	5 (11.4%)	Neomycin	1 (2.3%)
Chlortetracycline	2 (4.5%)	Polymyxin/neomycin sulfate/gramicidin	0 (0.0%)
Marbofloxacin	2 (4.5%)	Other antibiotic	4 (9.1%)
Florfenicol	1 (2.3%)	Unknown	10 (22.7%)

**Table 5 antibiotics-13-00382-t005:** Comparison of dogs and cats regarding feasibility of antibiotic administration using a 5-point Likert scale.

	**Overall** **n = 613**	**Dogs** **n = 409**	**Cats** **n = 204**	***p*-Value ^1^**	**q-Value ^2^**	**Cramer’s V ^3^**	**95% CI ^4^**
**How well were you able to give your animal the antibiotic?**				<0.001	<0.001	0.26	0.18–1.00
Not at all	4 (0.7%)	0 (0.0%)	4 (2.0%)				
Rather less good	23 (3.8%)	9 (2.2%)	14 (6.9%)				
Neutral	77 (12.6%)	34 (8.3%)	43 (21.1%)				
Rather good	99 (16.2%)	61 (14.9%)	38 (18.6%)				
Very good	410 (66.9%)	305 (74.6%)	105 (51.5%)				
	**Overall** **n = 44**	**Dogs** **n = 32**	**Cats** **n = 12**	***p*-Value ^1^**	**q-Value ^2^**	**Cramer’s V ^3^**	**95% CI ^4^**
**How well were you able to apply the antibiotic topically to your animal?**				0.089	0.089	0.29	0.00–1.00
Not at all	1 (2.3%)	1 (3.1%)	0 (0.0%)				
Rather less good	4 (9.1%)	1 (3.1%)	3 (25.0%)				
Neutral	8 (18.2%)	6 (18.8%)	2 (16.7%)				
Rather good	10 (22.7%)	6 (18.8%)	4 (33.3%)				
Very good	21 (47.7%)	18 (56.2%)	3 (25.0%)				

^1^ Fisher’s exact test; ^2^ False discovery rate correction for multiple testing with Benjamini & Hochberg method; ^3^ Interpretation: >0.25 very strong, >0.15 strong, >0.10 moderate, >0.05 weak, >0 no or very weak association [27]; ^4^ CI = confidence interval of effect size.

**Table 6 antibiotics-13-00382-t006:** Comparison of mode of oral antibiotic administration in dogs and cats, multiple answers were possible.

	Overall n = 584	Dogs n = 404	Cats n = 180	*p*-Value ^1^	q-Value ^2^	Cramer’s V ^3^	95% CI ^4^
Hidden in food/treats	264 (45.2%)	192 (47.5%)	72 (40.0%)	0.092	0.105	0.06	0.00–1.00
With food	201 (34.4%)	165 (40.8%)	36 (20.0%)	<0.001	<0.001	0.20	0.13–1.00
Given directly into the mouth	151 (25.9%)	86 (21.3%)	65 (36.1%)	<0.001	<0.001	0.15	0.08–1.00
Eating the tablets/capsules out of the hand without any additional aids	90 (15.4%)	69 (17.1%)	21 (11.7%)	0.094	0.105	0.06	0.00–1.00
Crushed tablets to powder and mixed with food/treats	39 (6.7%)	14 (3.5%)	25 (13.9%)	<0.001	<0.001	0.19	0.12–1.00
With tablet dispenser	18 (3.1%)	1 (0.2%)	17 (9.4%)	<0.001	<0.001	0.24	0.17–1.00
Crushed tablets to powder and injected into mouth with liquid via syringe	18 (3.1%)	4 (1.0%)	14 (7.8%)	<0.001	<0.001	0.18	0.11–1.00
With special treat, trojaner	17 (2.9%)	3 (0.7%)	14 (7.8%)	<0.001	<0.001	0.19	0.12–1.00
Capsule content given without capsule	2 (0.3%)	0 (0.0%)	2 (1.1%)	0.095	0.105	0.08	0.00–1.00
Other	12 (2.1%)	7 (1.7%)	5 (2.8%)	0.528	0.528	0.00	0.00–1.00

^1^ Pearson’s Chi-squared test; Fisher’s exact test; ^2^ False discovery rate correction for multiple testing with Benjamini & Hochberg method; ^3^ Interpretation: >0.25 very strong, >0.15 strong, >0.10 moderate, >0.05 weak, >0 no or very weak association [27]; ^4^ CI = confidence interval of effect size.

## Data Availability

The original contributions presented in the study are included in the article/Appendix A, further inquiries can be directed to the corresponding author/s. The raw data supporting the conclusions of this article will be made available by the authors on request.

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
