# Peer review of "Online-Assisted Survey on Antibiotic Use by Pet Owners in Dogs and Cats"

_antibiotics, 2024, doi:10.3390/antibiotics13050382_

Round 1
Reviewer 1 Report
Comments and Suggestions for Authors
The authors are advised to submit a supplementary document mentioning a list of case history to backtrack antibiotics used, mode of antibiotics applied, and disease condition of respective dogs and cats. The authors are also encouraged to comment on relation (if any) between antibiotic administration and disease type.
Author Response
|
Response to Reviewer 1 Comments
|
||
|
1. Summary |
|
|
|
Thank you very much for taking the time to review this manuscript. Please find the detailed responses below and the corresponding revisions/corrections highlighted/in track changes in the re-submitted files.
|
||
|
2. Questions for General Evaluation |
Reviewer’s Evaluation |
Response and Revisions |
|
Does the introduction provide sufficient background and include all relevant references? |
Yes |
|
|
Are all the cited references relevant to the research? |
Yes |
|
|
Is the research design appropriate? |
Yes |
|
|
Are the methods adequately described? |
Yes |
|
|
Are the results clearly presented? |
Yes |
|
|
Are the conclusions supported by the results?
|
Yes |
|
|
3. Point-by-point response to Comments and Suggestions for Authors
|
||
|
Comments 1: The authors are advised to submit a supplementary document mentioning a list of case history to backtrack antibiotics used, mode of antibiotics applied, and disease condition of respective dogs and cats.
|
||
|
Response 1: Thank you for your comment. As this was an anonymous questionnaire and detailed medical history information, such as the exact extent of symptoms and examination findings, as well as previous illnesses, could not be collected, it is not possible for us to provide an exact medical history for each dog and cat. The complete data available can be accessed upon request.
|
||
|
Comments 2: The authors are also encouraged to comment on relation (if any) between antibiotic administration and disease type.
|
||
|
Response 2: Thank you for your suggestion. We have created a supplementary table to provide an overview of the systemic antibiotic administration for the different indications. The table can be found in Supplementary Material Table S7. We have added the following sentence as a reference to the table in Line 202-204: “An overview on the selection of systemic antibiotics in regard to the disease process can be found in Table S7 in the Supplementary Material.” |
||
Reviewer 2 Report
Comments and Suggestions for Authors
First, the topic is interesting to do a survey on antibiotic use and to understand the recognition of the prudent use of antibiotics in pet owners' mind.
The manuscript is written well and easy to follow it.
The methodology is well designed and approved by the ethical committee.
Some minor points would like to ask for editing.
Line 113: One hundred and fifteen
Table 3: It would be better to show as an antibiotic class such as
Fluoroquinolones:
Enrofloxacin .......
Marbofloxacin .....
Author Response
|
Response to Reviewer 2 Comments
|
||
|
1. Summary |
|
|
|
Thank you very much for taking the time to review this manuscript. Please find the detailed responses below and the corresponding revisions/corrections highlighted/in track changes in the re-submitted files.
|
||
|
2. Questions for General Evaluation |
Reviewer’s Evaluation |
Response and Revisions |
|
Does the introduction provide sufficient background and include all relevant references? |
Yes |
|
|
Are all the cited references relevant to the research? |
Yes |
|
|
Is the research design appropriate? |
Yes |
|
|
Are the methods adequately described? |
Yes |
|
|
Are the results clearly presented? |
Yes |
|
|
Are the conclusions supported by the results? |
Yes |
|
|
3. Point-by-point response to Comments and Suggestions for Authors
|
||
|
Comments 1: Line 113: One hundred and fifteen
|
||
|
Response 1: Thank you for your correction, we have changed this in the manuscript (Line 123).
|
||
|
Comments 2: Table 3: It would be better to show as an antibiotic class such as
Fluoroquinolones: Enrofloxacin ....... Marbofloxacin ....
Response 2: Thank you for your suggestion, we have adjusted Table 3 accordingly.
|
||
Reviewer 3 Report
Comments and Suggestions for Authors
It was a very interesting study, especially knowing about some differences in Germany and other countries. The study provides good insight about the antibiotic administration, type of antibiotics used, form of application, tests carried out beforehand and factors influencing antibiotic prescription in Germany.
Author Response
|
Response to Reviewer 3 Comments
|
||
|
1. Summary |
|
|
|
Thank you very much for taking the time to review this manuscript. We are pleased that you found our study interesting. Please let us know if you have any specific comments on how we can improve the methodology presented.
|
||
|
2. Questions for General Evaluation |
Reviewer’s Evaluation |
Response and Revisions |
|
Does the introduction provide sufficient background and include all relevant references? |
Yes |
|
|
Are all the cited references relevant to the research? |
Yes |
|
|
Is the research design appropriate? |
Yes |
|
|
Are the methods adequately described? |
Can be improved |
|
|
Are the results clearly presented? |
Yes |
|
|
Are the conclusions supported by the results? |
Yes |
|
Reviewer 4 Report
Comments and Suggestions for Authors
While the study closes several research gaps in the field of antibiotic use in animals, the writing style suffers from verbosity and lacks coherence. Many sections could benefit from conciseness, as the purpose and key points of the argument are often unclear. Targeting at improving prudent antibiotic use and reducing antibiotic resistance (AMR), results should be focused on factors relevant to this goal. Additionally, the organization of the background section could be improved for better clarity and structure. Specific comments are listed below.
1. It is stated in the Abstract and Introduction that the aim of the study was to collect data from dog and cat owners on antibiotic administration, and the authors conducted a lot of comparisons between dog and cat owners. However, these repeated comparisons appear to lack significance for either research or policy implications, resembling unnecessary workload stacking. A more meaningful analysis would involve investigating the influencing factors behind inappropriate antibiotic use behaviors, such as how owners' attitudes affect antibiotic use for conditions like diarrhea, providing valuable insights for future interventions like educational campaigns. Presenting all descriptive results and merely comparing behaviors between dogs and cats seems to render the study as more of a basic report than a focused inquiry. It's regrettable that the authors didn't delve deeper into the rich dataset they collected.
2. The fifth paragraph of the Introduction is hard to understand due to confusing logic in the writing. While most of the paragraph discusses antibiotic over-prescription, the final sentence appears to convey an opposing viewpoint, creating inconsistency.
3. Similarly, the sixth paragraph of the Introduction is confusing as it primarily lists evidence from previous studies without clearly establishing the rationale behind the current study or its necessity.
4. On line 105, it's noted that 29.3% of respondents had medical training, a percentage higher than that of the genuine study population. This discrepancy could significantly bias the study results, yet the authors failed to provide any explanation for it.
5. The writing should be improved and require editing. For example, Line 116: “The median total age was 5 years for both dogs (IQR 3-9) and cats (IQR 3-10) between 1 and 18 years of age (p = 0.4).” Line 113: “Onehundredandfifteen”. Table 5: “How well were you able to give your animal the antibiotic?”
6. The questionnaire items presented in Figure 1 were not well-designed. Tendencies and biased language such as "always," "strictly," or "exactly" should be avoided, especially since the authors already utilized a 5-Likert scale for responses. It appears that there was a high degree of social desirability among the respondents.
7. Similarly, the questionnaire items presented in Figure 3 were also not well-designed. The introduction of the concept of AMR in the information sheet and the mention of reserve antibiotics in the corresponding question inadvertently educate respondents during the survey. To mitigate this effect, a screening question should have been included to ascertain whether respondents understood the concepts of AMR and reserve antibiotics before exploring their attitudes. This would help prevent the questionnaire itself from influencing respondents and their answers.
8. The discussion section is excessively lengthy and risks losing coherence. For instance, the paragraph spanning from Line 413 to 422 is convoluted and lacks clarity. It's important to avoid turning the article into a thesis, and instead, focus on succinctly conveying key points and insights.
9. Both “efficacy” and “effectiveness” are mentioned in the paragraph from Line 506 to Line 525. Make sure to understand the differences between the two words and use the appropriate one accordingly.
10. Line 591 to 593. Whether to choose Fisher's exact test depends on whether the theoretical frequency is less than 5, not the sample size.
Comments on the Quality of English LanguageThe writing should be improved and require editing. For example, Line 116: “The median total age was 5 years for both dogs (IQR 3-9) and cats (IQR 3-10) between 1 and 18 years of age (p = 0.4).” Line 113: “Onehundredandfifteen”. Table 5: “How well were you able to give your animal the antibiotic?”
Author Response
|
Response to Reviewer 4 Comments
|
||
|
1. Summary |
|
|
|
Thank you very much for taking the time to review this manuscript and the valuable comments and suggestions, which were very helpful for improving the manuscript. Please find the detailed responses below and the corresponding revisions/corrections highlighted/in track changes in the re-submitted files.
|
||
|
2. Questions for General Evaluation |
Reviewer’s Evaluation |
Response and Revisions |
|
Does the introduction provide sufficient background and include all relevant references? |
Can be improved |
|
|
Are all the cited references relevant to the research? |
Yes |
|
|
Is the research design appropriate? |
Can be improved |
|
|
Are the methods adequately described? |
Can be improved |
|
|
Are the results clearly presented? |
Can be improved |
|
|
Are the conclusions supported by the results?
|
Not applicable |
|
|
3. Point-by-point response to Comments and Suggestions for Authors
|
||
|
Comments 1: It is stated in the Abstract and Introduction that the aim of the study was to collect data from dog and cat owners on antibiotic administration, and the authors conducted a lot of comparisons between dog and cat owners. However, these repeated comparisons appear to lack significance for either research or policy implications, resembling unnecessary workload stacking. A more meaningful analysis would involve investigating the influencing factors behind inappropriate antibiotic use behaviors, such as how owners' attitudes affect antibiotic use for conditions like diarrhea, providing valuable insights for future interventions like educational campaigns. Presenting all descriptive results and merely comparing behaviors between dogs and cats seems to render the study as more of a basic report than a focused inquiry. It's regrettable that the authors didn't delve deeper into the rich dataset they collected.
|
||
|
Response 1: Thank you for your comment. Due to the overall lack of data on the use of antibiotics in dogs and cats in Germany, we also wanted to collect basic data first. As this was an anonymous survey and we have no specific data regarding the exact medical history of the dogs and cats and therefore justification for antibiotic administration (e.g. septicemia), this would be an interesting point for future research. With regard to your comment about how owners' attitudes affect antibiotic use for conditions such as diarrhoea, we have now used our data to compare the reasons given for antibiotic use and the expectation of antibiotic use. The table was included in the Supplementary Materials Table S4. We have inserted the following sentences in the manuscript in relation to this data:
Page 4, Line 153-156: “The owner expectations for antibiotic administration for different indications are displayed in the Supplementary Material in Table S4. The owners' highest expectation of antibiotic administration was for castration (40/42; 95.2%), soft tissue surgery (64/69; 92.8%), and other surgery (11/12; 91.7%%).”
Page 14, Line 407-409: “When comparing the expectation to receive an antibiotic with the reason for administration, it is noticeable that most owners expected an antibiotic for castration (95.2%).”
Page 14, Line 413-415: “It is a striking result of the study that 81.9% of owners were expecting antibiotic therapy for their pet. In contrast, in a study in the UK, only 49.2% of cat owners expected antibiotics for treatment [26]. “
Page 17, Line 566-568: “Due to the high expectations of owners regarding the administration of antibiotics, owner education about prudent use of antimicrobials seems critically important.”
Page 17, Line 599-600: “Further research would be indicated in this regard in order to assess more clearly the factors influencing inadequate antibiotic prescription and the influence of pet owners. “
|
||
|
Comments 2: The fifth paragraph of the Introduction is hard to understand due to confusing logic in the writing. While most of the paragraph discusses antibiotic over-prescription, the final sentence appears to convey an opposing viewpoint, creating inconsistency.
|
||
|
Response 2: Thank you for your comment. This paragraph is also intended to show that there are different opinions and uncertainties regarding expectations. On the one hand, owners think that vets prescribe antibiotics too often and trust their vet to make a decision and would not be disappointed if veterinarians did not give an antibiotic. On the other hand, they would prefer an antibiotic to be given if it is unclear whether it is necessary at all.
We have changed the last sentence and tried to make this a little clearer: Page 2, Line 84-86: “On the other hand, the majority of pet owners stated that they would prefer to treat their pet with antimicrobial drugs even if the benefit of an antibiotic was not clear [21].”
Comments 3: Similarly, the sixth paragraph of the Introduction is confusing as it primarily lists evidence from previous studies without clearly establishing the rationale behind the current study or its necessity.
Response 3: Thank you very much for this comment. In this section we wanted to point out that cat owners in particular often have problems administering oral medication and that this can be a reason for using long-acting injections. As cefovecin is a 3rd generation cephalosporine and therefore an HPCIA, its use should be restricted and consideration should be given to using it simply because it is easy to administer. With this study we wanted to collect data on antibiotic use and in particular the frequency of use of cefovecin.
We have added the following sentence in Line 104-106, Page 3: “A special interest was in the use reported for the 3rd generation cephalosporine cefovecin as a critically important antibiotic.”
Comments 4: On line 105, it's noted that 29.3% of respondents had medical training, a percentage higher than that of the genuine study population. This discrepancy could significantly bias the study results, yet the authors failed to provide any explanation for it.
Response 4: Thank you very much for this important remark. For improvement, we have compared the following three points for owners with and without medical training:
- Comparison between owners with and without medical training and the expectation of antibiotic administration. The following sentence was added for this purpose:
- Comparison of medical training and the questions in Figure 1 with regard to information provided by the veterinarian and compliance with antibiotic administration. The data can be viewed in the Supplementary Materials in Table S8 and we have added the sentence in the manuscript:
- Comparison between owners with and without medical training and hand washing or wearing gloves when administering antibiotics. The data can be viewed in the Supplementary Materials in Table S9 and we have added to the manuscript:
We have also added a reference to a possible bias in the limitations:
Page 17, Line 587-588: “Another factor potentially influencing the results is that 29.3% of participating owners reported having received medical training. “
Comments 5: The writing should be improved and require editing. For example, Line 116: “The median total age was 5 years for both dogs (IQR 3-9) and cats (IQR 3-10) between 1 and 18 years of age (p = 0.4).” Line 113: “Onehundredandfifteen”. Table 5: “How well were you able to give your animal the antibiotic?”
Response 5: Thank you for the comment. Adaptions were made to improve language. These are highlighted in red in the manuscript. We have corrected “Onehundredandfifteen” to "One hundred and fifteen" (Line 123).
Comments 6: The questionnaire items presented in Figure 1 were not well-designed. Tendencies and biased language such as "always," "strictly," or "exactly" should be avoided, especially since the authors already utilized a 5-Likert scale for responses. It appears that there was a high degree of social desirability among the respondents.
Response 6: Thank you for your comment. As we are unable to change this retrospectively, however, we have addressed this point in the limitations.
Page 17, Line 588-592: “ In addition, the design of the questions might have influenced the answers of owners, using phrases such as 'always', 'strictly' or 'exactly', and mentioning the problems of antibiotic resistance, that could potentially have promoted answers owners appreciated as being more socially desirable.”
Comments 7: Similarly, the questionnaire items presented in Figure 3 were also not well-designed. The introduction of the concept of AMR in the information sheet and the mention of reserve antibiotics in the corresponding question inadvertently educate respondents during the survey. To mitigate this effect, a screening question should have been included to ascertain whether respondents understood the concepts of AMR and reserve antibiotics before exploring their attitudes. This would help prevent the questionnaire itself from influencing respondents and their answers.
Response 7: Thank you for your comment. We agree with you on this point and have added a corresponding comment in the limitations, as a change is retrospectively not possible.
Page 17, Line 588-592: “ In addition, the design of the questions might have influenced the answers of owners, using phrases such as 'always', 'strictly' or 'exactly', and mentioning the problems of antibiotic resistance, that could potentially have promoted answers owners appreciated as being more socially desirable.”
Comments 8: The discussion section is excessively lengthy and risks losing coherence. For instance, the paragraph spanning from Line 413 to 422 is convoluted and lacks clarity. It's important to avoid turning the article into a thesis, and instead, focus on succinctly conveying key points and insights.
Response 8: Thank you for this important suggestion for improvement. We have rewritten the relevant section (Line 413 to 422). The changes are visible by track changes in the manuscript and the current section reads as follows:
Page 14-15, Line 440-458: Surprisingly, metronidazole was indicated as the third most frequently used antibiotic in the present study. Metronidazole is a nitroimidazole and, unlike amoxicillin and amoxicillin/clavulanic acid, has a fairly narrow spectrum of activity against obligate anaerobes and protozoa like Giardia [53,54]. According to a study from the USA, metronidazole was the most commonly used antibiotic for gastrointestinal disorders in dogs and cats [30]. In the present study, diarrhea was also reported to be the most common reason for the administration of metronidazole. However, this should be discussed critically, as metronidazole is known to cause significant and longstanding dysbiosis of the enteral microbiome [39] and in most cases of acute and chronic diarrhea, antibiotic treatment does not affect clinical improvement in stable patients [31-38, 55, 56]. For treatment of Giardiasis, fenbendazole has been recommended as the first-line drug, causing only minimal changes in the enteral microbiota while being equally effective against Giardia [33, 37, 57-59].
We also reviewed the length of the discussion and shortened it. The deleted sentences are still visible by track changes. |
||
|
|
||
|
Comments 9: Both “efficacy” and “effectiveness” are mentioned in the paragraph from Line 506 to Line 525. Make sure to understand the differences between the two words and use the appropriate one accordingly.
Response 9: Thank you very much for pointing this out to us. We have reviewed definitions of both terms and have decided to use "efficacy" for our collected data and have retained "effectiveness" in our manuscript, in accordance with the terminology employed by the authors of the corresponding papers.
Comments 10: Line 591 to 593. Whether to choose Fisher's exact test depends on whether the theoretical frequency is less than 5, not the sample size.
Response 10: Thank you for the correction, we have adjusted this accordingly in the manuscript.
Page 18, Line 632-635: “All data have been analyzed descriptively, and statistical comparisons between groups in categorical variables have been performed using the Chi-squared test in samples > 5, while the Fisher`s exact test was used, if the theoretical frequency was less than five.”
|
||
|
4. Response to Comments on the Quality of English Language
|
||
|
Point 1: Line 116: “The median total age was 5 years for both dogs (IQR 3-9) and cats (IQR 3-10) between 1 and 18 years of age (p = 0.4).” |
||
|
Response 1: I appreciate your feedback. Language editing was performed by a native speaker and the manuscript was revised accordingly. As no grammatical errors were identified in this sentence, this sentence was not modified. You are welcome to make specific suggestions for improvement.
Point 2: Line 113: “Onehundredandfifteen” Response 2: Thank you for pointing this out. We have corrected this in the manuscript (Line 123).
Point 3: Table 5: “How well were you able to give your animal the antibiotic?” Response 3: Thank you for pointing this out. Since no grammatical errors were found in this sentence during the language editing, and since it is American English, the sentence was not changed. If you have specific suggestions for improvement, please let us know.
|
||
|
5. Additional clarifications |
||
|
We have replaced the graphics shown in the manuscript with graphics in a resolution of 300dpi. Apart from the described language improvement of the three items in graphic 1, nothing has changed in the graphics themselves.
|
||
Round 2
Reviewer 4 Report
Comments and Suggestions for Authors
The manuscript has been enhanced overall following the revision. Thank you for your efforts.